# Time of Return to Work (RTW) May Not Correlate with Patient-Reported Outcomes Measurements (PROM) at Minimum One Year Post Arthroscopic Bankart Repair

**DOI:** 10.3390/jcm12185794

**Published:** 2023-09-06

**Authors:** Mateusz Kosior, Aleksandra Sibilska, Marcin Piwnik, Andrzej Borowski, Szymon Prusaczyk, Jason Rogers, Sławomir Struzik, Adam Kwapisz

**Affiliations:** 1Clinic of Orthopedics and Pediatric Orthopedics, Medical University of Lodz, 92-213 Lodz, Poland; mateusz.kosior@gmail.com (M.K.);; 2Department of Orthopaedics and Traumatology of the Musculoskeletal System, Infant Jesus Teaching Hospital, Medical University of Warsaw, 02-091 Warsaw, Polandsstruzik@wp.pl (S.S.); 3Ortoteam Clinic, 90-127 Lodz, Poland; 4Department of Orthopedics and Traumatology, Radomsko Community Hospital, 97-500 Radomsko, Poland; 5EmergeOrtho Triad Region, Greeensboro, NC 27408, USA; jrog25@yahoo.com

**Keywords:** shoulder dislocation, Bankart repair, return to work, shoulder stabilization, shoulder arthroscopy

## Abstract

It is widely recognized that work serves a dual role by not only ensuring financial independence but also functioning as a vital source of psychosocial well-being and contributing significantly to the attribution of meaning in life. The cost of work disability can be a multifactorial problem for both employers and workers; thus the inability to return to work (RTW) may have a destructive effect on mental health and confidence. Shoulder surgery is one of the conditions that inevitably impacts patients’ ability to work. As current data focus on restoring range of motion, strength, and the patients’ activity, to this day the data about RTW post shoulder surgery remain limited. The purpose of this study was to evaluate the return-to-work time of patients treated with an arthroscopic Bankart repair and to evaluate if patient-reported outcomes (PROM) correlate with the incapacity to work after an arthroscopic Bankart repair. We performed a retrospective review by conducting a questionnaire with patients more than 12 months after surgery and we identified 31 patients who met the criteria for the study and were able to contact 17 of them. In this paper we demonstrated that on average among groups working physically and at the office we may expect patients who underwent arthroscopic Bankart repair to return to work within 7 weeks from the surgery, with office workers tending to return significantly faster with an average of 2.5 weeks (*p* = 0.0239).

## 1. Introduction

Since most adults spend the majority of their lifetime at work, it is somehow crucial to identify work-related issues that can influence employees’ well-being [1]. One such factor is job insecurity. Two types of such have been described, objective and subjective job insecurity. Both are deeply interconnected. Job insecurity can lead to psychological distress and somatic issues; it can affect mental as well as physical health [2].

Uncertainty of returning to work may affect the way employees perceive their future life, and that can affect the capacity to cope with daily life endeavors as well as having a substantial impact on the family well-being [2]. The inability to work can have a detrimental effect on both the macro and micro economy. Work not only provides financial independence but is a source of psychosocial well-being and is an important provider of meaning. Therefore, the inability to return to work (RTW) may have a detrimental effect on mental health and self-esteem [2,3].

Shoulder surgery, as with any other surgical procedure, is one of the conditions that inevitably impacts patients’ lives. Current data focus on restoring the range of motion, strength, and the patients’ activity. Bankart repair is one of the common procedures among shoulder surgeries, pioneered by Bankart in 1923 and still being innovated by another generation of shoulder surgeons [4]. According to the American Board of Orthopaedic Surgery, from 2003 to 2008 a total of 4562 Bankart repair cases were reported, comprising 8.6% of the total number of shoulder surgery cases. It is also interesting to note that the number of surgeons performing arthroscopic Bankart repairs increased during the period of study from 123 in 2003 to 217 in 2008 [5]. Given that these figures are from quite a number of years ago, we can estimate that the amount of such procedures has been multiplied. With procedures being performed as frequently around the world, the eventual complications described will affect a significant number of patients. It has been reported that Bankart repair can lead to up to 11% of failures; however, with a longer follow up, the instability rate can rise to 31%. A return to sport after an arthroscopic Bankart repair can be achieved even in 77% of cases. There are also data reporting that osteoarthritis (OA) can be expected in 31% of patients who underwent Bankart repair [6]. There are also some studies with a longer follow-up (mean follow-up of 55.93 months) where, in addition to Bankart repair, remplissage was performed and they noted in total a recurrence of instability in 2 trauma cases (14.28%) [7]. However, previously mentioned issues were deeply analyzed, and to this day the data about RTW post shoulder surgery still remain limited.

To the best of our knowledge, the biggest available study reports 1773 cases, with most patients treated arthroscopically by a single shoulder surgeon. In this retrospective review, the authors included patients who underwent primary shoulder surgery such as: acromioplasty for rotator cuff impingement, Bankart repair and superior labral anterior to posterior (SLAP) repair for labral tears/instability, calcific debridement for calcific tendonitis, arthroscopic capsular release for idiopathic adhesive capsulitis, arthroscopic rotator cuff repair for rotator cuff tears and synthetic polytetrafluoroethylene patch rotator cuff repair for massive irreparable rotator cuff tears. Open surgeries were also performed such as anatomical total shoulder arthroplasty, hemiarthroplasty and reverse shoulder arthroplasty. Grouping in this study was according to the type of the surgery and more than 10 patients were required to have undergone a specific type of procedure to include procedure in the study. Patients were evaluated at 1, 6 and 12 weeks preoperatively and 6 months postoperatively and the results were differentiated by type of procedure and for all procedures combined. The average age for all procedures was 55 ± 0.3 (range 12–91) years.

It is reported that overall, 77% patients managed to RTW at 6 months, yet patients who had undergone arthroscopic procedures were statistically more prone to return to their duties [8]. It is worth adding a note that in their study concomitant rotator cuff repair with stabilization and Bankart repair were performed in comparatively young cohorts with a mean age of, in sequence, 42 and 28. This may, naturally, have an impact on the RTW rates [8].

As for the group of patients after Bankart repair, which we also examined, Jayasekara et al. described that arthroscopic Bankart repair (ABR) gives an 84% chance of a return to work, with a 51% chance of a return to full duties within 6 months of the surgery [8]. In a different paper, Ateschrang et al. verified the inability to work after open and arthroscopic Bankart repair for post-traumatic anterior instability of the shoulder joint without features of hyperlaxity. They examined 93 patients with a mean age of 37.1 (Standard Deviation (SD) ± 14.4) and with a mean follow-up of 48.3 months (SD ± 23.6) and concluded that the mean incapacity for work in the group of arthroscopic Bankart repair was 3.3 months and 2.7 months in the group of open Bankart repair (they noted no statistical difference, with *p* = 0.37). Interestingly, it was also found that the average time to return to work after arthroscopic Bankart repair for patients with a low workload was 2.4 months and 4.2 months for those with a high workload [9].

Another study, conducted at the same facility as the one cited above, carried out by Kraus et al., checked patient’s incapacity to work after Bankart repair according to the classification by the REFA (the Association for Work Design, Business Organization and Business Development) (Darmstadt, Germany). They also measured recovery time and the outcome of patients with a heavy workload was compared to those with lower workloads. A total of 74 patients met the inclusion criteria with the mean age of 34.7 years (SD ± 12.6) and mean follow-up time of 43.1 months (SD ± 17.4). They found that the mean incapacity to work was 2.73 months overall. Subdividing this according to the REFA classification, the incapacity to work was 2.06 months in the group with low physical strains at work (REFA 0–1) and 3.40 months in the group with a heavy workload (REFA 2–4) [10].

Therefore, the purpose of this study is to evaluate the return-to-work time of patients treated with an arthroscopic Bankart repair procedure. The secondary aim is to evaluate whether patient-reported outcomes (PROM) correlate with the incapacity to work after an arthroscopic Bankart repair procedure.

## 2. Material and Methods

We obtained the approval of the Bioethical Commission of the Medical University of Lodz prior to commencing our study (RNN/125/23/KE, approved on 16 May 2023).

A retrospective review of the hospital’s database was performed to identify all the patients who underwent arthroscopic Bankart repair. Next, we excluded patients who did not match inclusion criteria. Inclusion criteria were: 1. Age over 17 years; 2. Follow-up of at least 12 months from the surgery; 3. No prior surgery of the affected shoulder; 4. Either email or phone number available in the hospital’s database; 5. Surgery performed by the fellowship-trained shoulder surgeon. Exclusion criteria were: 1. Patient not working or retired prior to surgery; 2. Previous surgeries of the affected shoulder; 3. Surgery performed by other surgeons than the senior author; 4. Open Bankart procedure; 5. Bone procedure as for example Latarjet; 6. Follow-up of less than 1 year; 7. Lack of the phone number or email in the hospital database; 8. History of fractures around the affected shoulders. The surgical procedures were conducted by a shoulder surgeon who had received specialized training through a world-recognized shoulder surgery and sports orthopedics fellowship program. This surgeon commenced their practice at our medical institution in 2019; therefore, the scope of our database search was confined to the period spanning 2019 to 2022. We also excluded all the patients who were either retired or unable to work for a reason other than shoulder surgery from our further analysis.

Further, each included patient was attempted to be contacted via phone or email; those contacted via email were asked to provide their phone number for further proceedings. Those who responded were asked to provide the time needed to return to work, as well as the type of work, either physical or an office job. They were also surveyed by means of Single Assessment Numeric Evaluation (SANE), Global Rating of Change (GRoC) and Simple Shoulder Test (SST) scoring systems.

SANE was introduced by Williams et al. in 1999 as a simple, single-question evaluation: “How would you rate your shoulder today as a percentage of normal (0% to 100% scale with 100% being normal)?” [11]. It was later independently validated in shoulder patients [11].

GRoC is a questionnaire that provides a measure of global well-being that is based on progress, or its lack, since an initial treatment [12,13,14]. The scales are very widely used both in clinical practice and research settings, especially in the musculoskeletal area.

SST is a shoulder-specific questionnaire that measures functional limitations of the affected shoulder in patients with shoulder dysfunction. SST consists of 12 questions, with a response of yes/no, evaluating patients’ tolerance for specific activities [15]. All the questionnaires described above and utilized in the study, along with their corresponding questions, are shown in Table 1.

Statistical analysis was performed using Statistica 13.3 (TIBCO Software Inc., Palo Alto, CA, USA). The correlation between numerical variables was examined using the non-parametric Spearman correlation. Average values in two groups were compared using the Mann–Whitney U test. All the average values are reported as a median with min–max range. The level of statistical significance in all tests was set at the level α = 0.05.

## 3. Results

In the hospital database we identified 31 patients eligible for our study, of whom we managed to contact 17 patients via phone: the studied group consisted of 4 females and 13 males. The average patient’s age at the day of surgery was 30 years (range 17–57) and the follow-up on the day of the survey ranged from 12 to 49 months.

In our survey we asked our patients whether they managed to return to work (RTW) or not. All but one of the patients managed to return to work, which resulted in a 94% ratio of return to work. The only unsuccessful patient had complications due to COVID-19 infection; therefore, he could not return to his work duties. None of the complications were related to the shoulder surgery.

The study’s patients managed to return to work at a median of 7 weeks, with the return-to-work period ranging from 1 week to 20 weeks.

The time of returning to work substantially varied depending on the type of work performed. Office workers returned significantly more quickly to work after an average of 2.5 weeks (range 1–17), so on average within less than 1 month after the surgery, while patients performing physical work returned to work after 12 weeks (4–20), so on average within 3 months after the Bankart repair. The difference between the groups was statistically significant, *p* = 0.0239 (Table 2).

At the last follow-up visit, the average SANE score was 90 (range 80–100), GRoC score 6 (range 0–7) and SST 12 (7–12), respectively. All of these parameters proved to have good to excellent outcomes one year after the surgery. Spearman’s rank correlation was computed to assess the relationship of RTW time with SANE and GRoC. There was no correlation between any of these pairs; r(14) = −0.41, *p* = 0.1164 for RTW vs. SANE and r(14) = −0.16, *p* = 0.5643 for RTW vs. GRoC. Computation of the correlation coefficient between RTW and SST was impossible, as all patients without missing RTW data achieved a result equal to 12 in SST.

We also stratified the results regarding the type of work. Patients with office work did not differ from physical workers in terms of scores on the SANE, GRoC and SST scales. All of the results are summarized in Table 3 and Table 4.

## 4. Discussion

To our knowledge this is the first study, based on arthroscopic Bankart repair patients, that not only evaluated the time of return to work, but tried to correlate RTW with patients-reported outcomes. In this study, patients tended to return to work on average within seven weeks post-surgery, which has no correlation with PROMs measured one year after the surgery.

We can define the success of shoulder surgery through various determinants. We can evaluate the failure rate, post operative range of motion, patient individual satisfaction or how many of our patients could return to their hobby. But if we understand that most of our patients will return to work and spend the majority of their time at work, RTW may be one of the most important, yet underestimated, indexes.

Hasselkus and Rosa stated that work itself is an important provider of meaning for the worker [16]. According to Waddell and Burton, work is healthy as it can provide not only financial independence but also psychosocial well-being. It can also be a source of identity and social status [17]. Therefore, job insecurity is one of the aspects that has come under the spotlight in the last few decades. As mentioned before, this factor has even been divided into two types, objective and subjective. The second type includes situations in which employees can be in fear of discontinuation of their job or even possible loss of such. This can result in psychological and mental distress, and even lead to a somatic manifestation [18,19]. Job insecurity is a substantial stressor that can affect not only one’s health, but also perception of the future of an employee’s family [2].

Furthermore, health conditions resulting in an inability to work can have a gross impact on the global economy, and according to the World Bank and the World Health Organization can cost trillions of US dollars yearly [20]. For individual workers, the inability to work may result in a poorer physical and mental condition. If unable to work, people must develop a new identification process, which can impact the meaning of their role in society [2,3].

If we take a closer look at the data, we can assume that, depending on the technique, stability can be restored in up to 91% of cases; however, arthroscopic repair can result in more failures. Historically, it was reported that as many as 23–49% patients could have redislocation; however, with the evolution of the techniques these numbers dropped gradually to 8–11% [6]. Murphy et al. conducted a systematic review of the long-term outcomes post arthroscopic Bankart repair. They analyzed 822 shoulders in 9 studies, reporting that, at the mean follow-up of 12 years, as many as 31% of patients may expect incidences of instability. What can be more frustrating is that OA changes were even reported in 59% of arthroscopically treated cases [21].

One of the extensively tested indexes is the return to sport after shoulder surgery. AlSomali in his systematic review, based on 11 papers which included 566 shoulders from 563 patients, found that a return to sport can be achieved even in 87% of patients who underwent an open Bankart repair [6]. Gouveia et al. reviewed 20 studies, with reports of, in total, 738 shoulders of 736 patients. Interestingly, 83% of these patients were male, with a mean age of 28 years (range 14–72 years) and a mean follow-up time after surgery of 45 months (range 12–127 months). He found that the return to sport ranged from 60 to 100% of cases with differences in terms of the type of sport. For contact or collision athletes it ranged from 80% to 100%, whereas for overhead or throwing athletes it was 46% to 79%. The rate of recurrence of instability at the maximum was 20% among the papers analyzed [22]. These findings regarding the return to sport are also at similar levels to those presented in the literature after other frequent arthroscopic shoulder surgeries like rotator cuff repair [23].

Currently most available studies report surgically oriented measures as a range of motion, complication rate or strength, leaving data about the impact of surgery on peoples’ way of life behind. To our knowledge, the biggest study, yet still one of just a few, consists of 1773 consecutive patients who underwent shoulder surgeries of different techniques. The authors concluded that 4 out of 5 patients were able to RTW within six months after surgery, yet about only half of them could return to full duty [8].

The Bankart repair is thought to be one of the most common methods of surgical treatment of an unstable shoulder [24]. Regardless of its popularity, we have limited resources to discuss the return-to-work time with our patients. Jayasekara et al. reported that 84% of patients were able to return to work [8]. In our study, the RTW ratio was 94%, yet the only unsuccessful patient could not return to his duties due to suffering from COVID-19 complications. The high rate of RTW is also supported by Kraus et al., who reported that only 4% of their patients had to change their job after the BR, with only 17.5% patients reporting some difficulties [9]. However, in the Jayaskara et al. paper, only 51% of treated people could return to their full duties [8].

According to Kraus et al., the mean incapacity to work was 2.73 months; however, this period varies depending on the age of the patients and is the shortest for people with an age range from 30 to 45. For such patients, one can expect RTW within 2.08 months. A longer RTW has been reported by Ateschrang et al., for whose patients it took an average of 3.3 months. It should be noted, however, that both studies were performed by the same authors [9,10]. The average RTW time for our patients was 8.25 weeks. If we convert that into months, it results as 2.06 months and approximates us to Kraus’s study. Noticeably, those who had an office job experienced a faster RTW; however, the difference was far from being significant.

In the Kraus study, at the last follow-up, patients reported 87.6 ROWE Score points and 87.7 Constant and Murley Score points, which can be interpreted as a good or even excellent result of the Bankart repair [10,25]. The same tools were used by Ateschrang et al.; however, there is a high chance cases overlapped [9]. Our patients also revealed high PROM at the last follow-up, with their SANE exceeding 90%, SST over 11 and GRoC close to 6. However, to our knowledge we are the first to try and correlate PROMs with the RTW. Unfortunately, we could not detect any significant correlation for any of these scoring systems. What we managed to detect was a slightly higher SANE score for office work, but that can be justified with lower workloads of such a job versus more physical duties.

This study is not without limitations. Unfortunately for our study, in our community, arthroscopic Bankart repair seems to be offered less often than a bone procedure such as the Latarjet technique. We also managed to contact only 54.8% of eligible patients, and regarding the fact that our shoulder surgeon started his practice in 2019, that resulted in a limited number of cases analyzed in this study. Nevertheless, to our knowledge this is the first attempt to evaluate if patient-reported outcome measurements correlate with the time needed to return to work after the ABR.

## 5. Conclusions

This study reports that patients who underwent arthroscopic Bankart repair may expect to return to work within 7 weeks, meaning less than 3 months, from the surgery, with office workers tending to return significantly faster. Those may require on average 2.5 weeks before returning to work. Patient-reported outcomes may not correlate with the time needed to return to job duties.

## Figures and Tables

**Table 1 jcm-12-05794-t001:** Presentation of all three questionnaires used in the study.

GRoC	How would you rate your shoulder in comparison to the situation before the surgery, if −7 is much worse and 7 is much better?
SANE	How would you rate your shoulder on the scale from 0 to 100, if 100 is a painless shoulder with full functionality?
SST	1. Is your shoulder comfortable with your arm at rest by your side? 2. Does your shoulder allow you to sleep at night? 3. Can you reach the small of your back to tuck in your shirt with your hand? 4. Can you place your hand behind your head with the elbow straight out to the side? 5. Can you place a coin on a shelf at the level of your shoulder without bending your elbow? 6. Can you lift 0.5 kg to the level of your shoulder without bending your elbow? 7. Can you lift 4 kg to the level of the top of your head without bending your elbow? 8. Can you carry 10 kg at your side with the affected extremity? 9. Do you think you can toss a tennis ball underhand 10 m with the affected extremity? 10. Do you think you can toss a tennis ball underhand 20 m with the affected extremity? 11. Can you wash the back of your opposite shoulder with the affected extremity? 12. Would your shoulder allow you to work full-time at your usual job?

**Table 2 jcm-12-05794-t002:** Summary of the most important findings.

	*n*	Median	1. Quartile	3. Quartile	Min	Max
age	17	30	24	34	17	57
RTW (weeks)	16	7	2.5	13	1	20
RTPA (months)	16	5	3	6.5	2	12
SANE (0–100)	17	90	85	95	80	100
GROC (−7–7)	17	6	6	7	0	7
SST (0–12)	17	12	12	12	7	12

(RTW—return to work, RTPA—return to physical activity).

**Table 3 jcm-12-05794-t003:** Brief summary of all analyzed correlations for the whole cohort.

	df	R	*p*
Age vs. RTW	14	0.57	0.0220
Age vs. RTW	14	−0.26	0.3274
Age vs. SANE	15	−0.11	0.6673
Age vs GRoC	15	0.12	0.6448
Age vs. SIMPLE SHOULDER TEST	15	−0.41	0.1021
RTW vs. RTPA	14	−0.03	0.9069
RTW vs. SANE	14	−0.41	0.1164
RTW vs. GRoC	14	−0.16	0.5643
RTPA vs. SANE	14	−0.07	0.8009
RTPA vs. GRoC	14	−0.18	0.5078
SANE vs. GRoC	15	0.55	0.0216
SANE vs. SIMPLE SHOULDER TEST	15	−0.34	0.1783
GROC vs. SIMPLE SHOULDER TEST	15	−0.27	0.2859

(RTW—return to work, RTPA—return to physical activity).

**Table 4 jcm-12-05794-t004:** Descriptive statistics of the group by type of work performed.

	Office Job			Physical Job			
	Median	Min	Max	Median	Min	Max	*p*
RTPA (weeks)	4.5	2	10	5.5	3	12	0.5635
RTW (weeks)	2.5	1	17	12	4	20	0.0207
SANE (0–100)	90	80	100	90	80	100	0.5286
GROC (−7–7)	6	0	7	6	3	7	0.9164
SIMPLE SHOULDER TEST (0–12)	12	12	12	12	12	12	0.9581

(RTW—return to work, RTPA—return to physical activity).

## Data Availability

Not applicable.

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
