# Peer review of "Time of Return to Work (RTW) May Not Correlate with Patient-Reported Outcomes Measurements (PROM) at Minimum One Year Post Arthroscopic Bankart Repair"

_jcm, 2023, doi:10.3390/jcm12185794_

Round 1

Reviewer 1 Report

Time of return to work (RTW) may not correlate with Patients Reported Outcomes Measurements (PROM) at minimum one year post arthroscopic Bankart repair

-Review-

General comments

​The present manuscript reports on valuable information regarding return to work following arthroscopic Bankart repair. It is a fact that such data is currently very limited, therefore the present study provides valuable insight into the cocept.

​In my opinion the Abstract and Introduction sections need to be restructured. The Abstract section needs to be written according to journal recommendations while the Introduction section should provide the reader with more concrete information on the concept of return-to-work following shoulder surgery and its many facets.

​There are small improvements that can be made in the Methods and Results section. Please see my suggestions.

​Please consider improving your Discussion section. I have provided some small suggestions that I hope you will find useful.

​Kind regards,

Abstract

​Please structure your abstract according to the journals template (https://www.mdpi.com/journal/jcm/instructions). In its present form the abstract does not follow journal recommendations.

​Line 15-16 – Please rephrase the introductory sentence to better fit an academic audience. My suggestion is: “It is widely recognized that work serves a dual role by not only ensuring financial independence but also functioning as a vital source of psychosocial well-being and contributing significantly to the attribution of meaning in life.

​Line 26-27 – Please also state the time that office workers require to return to work (line 26). Furthermore, if you use the word “significantly” please provide a p-value used to assess this.

Introduction

​Line 33 – 34 – It is my humble opinion that the Introduction section should not repeat phrases from the Abstract section. This is because each section has clear aims. Please use the Abstract section to convey the most relevant aspects of your work within the 200-word limit. Please use the Introduction section to provide the reader with the broad context of the topic while at the same time emphasizing the significance of your research question.

​Line 32 – 36 – In my opinion it would be better to start the introduction with a discussion of RTW after shoulder surgery. This might enhance brevity and clarity of your statements. This is only an opinion, not necessarily a suggestion.

​Line 37 – In my opinion this sentence does not convey any real meaning because it is too general. Any surgery or treatment impacts a patient’s life given the important burden of disease.

​Line 37 – 43 – Although interesting this sentence lacks a clear topic. Please also mention the reasons why patients underwent arthroscopic shoulder surgery. In my opinion RTW following Bankart repair might be different from rotator cuff repair. Furthermore, one should also consider the age of the patient as younger patients might have different views on work than older ones.

​Line 37 – 43 – “Most patients treated arthroscopically” – In my opinion it is difficult to interpret this sentence because RTW from arthroscopic surgery might be different from open surgery. Could you please comment on this in care you also agree that it is relevant?

​Line 44 – 48 – In my humble opinion this paragraph is better suited for the Discussion section. Furthermore it is my opinion that it can be reduced to a single sentence that has two concepts 1) RTW is a dynamic concept, many authors understand different things (RTW vs. return to full duties vs. incapacity of work) and 2) reported time-frames differ very much. In my opinion it would be better to further develop the concept of RTW following shoulder surgery so that the reader can better understand it. For example does open or arthroscopic surgery contribute in any way to RTW? What about different shoulder pathologies? What does the literate say about duration of surgery and RTW? Is there a link between RTW and postoperative range of motion?

Material and Methods

​Line 54 – 55 – Please consider rephrasing this section by starting with “Inclusion criteria were …” and “exclusion criteria were …”.

​Line 57 – 59 – In my opinion this section needs to be rephrased because it is not clear. I suggest “The surgical procedures were conducted by a shoulder surgeon who had received specialized training through a fellowship program. This surgeon commenced their practice at our medical institution in 2019; therefore, the scope of our database search was confined to the period spanning 2019 to 2022.

​Line 81 – 82 – Usually it is better to start the Methods section with a short sentence regarding the ethical approval of the study.

Results

​Line 86, 88 in conjunction with Line 78-79 – Please note that standard deviation is only valuable to describe the dispersion in normally distributed quantitative variables and in a case that the variable is not normally distributed another dispersion indicator called interquartile range (IQR= Q3-Q1) is used.

​Line 88 – Please correct “Study’s” to “study’s”.

​Line 90 – In this case I see that you reported minimum and maximum value as a measure of dispersion. Please consider using the same measure of dispersion throughout the manuscript. If your data is normally distributed use SD, if not use IQR or min and max values if data is “very” non-normal with many outliers.

​Line 90 – Furthermore, please use the same time measure months/weeks/year. In my opinion if you interchange them this will generate much confusion for the reader.

​Line 95 – Please report follow-up at the beginning of the results section. Follow-up can be considered “demographic” along with Age, gender, BMI, and anything else that is relevant for your study population that was collected to perform this study.

​Line 96 – “at the follow-up” – it is not clear to what follow-up are you referring to. Please clarify.

​Line 97 – “none of those values” – Please rephrase this sentence to better adhere to academic language.

​Line 98 – When reporting correlation results, please use the following phrasing for Spearman correlation: Spearman's rank correlation was computed to assess the relationship between [variable 1] and [variable 2]. There was a [negative or positive]/ [no] correlation between the two variables, r(df) = [r value], p = [p-value]

​Line 99 – “significantly higher SANE scores” – Please also mention in comparison to whom did patients with office jobs have better SANE scores. “Significantly higher …” compared to whom?

Discussion

​Line 110 – 113 – In my opinion I would very much like to see what the authors have to say about current literature reports on the subject and how their results compare to literature. It is my humble opinion that the authors should not repeat sentences from the Abstract (Line 15 -16) and Introduction section (Line 33 -34).

​Line 114 – 118 – Same comment as above, this paragraph is rephrasing the first part of the Introduction section (Line 32 -36). Please consider discussing your results in relation to available literature. 

​Line 120 – 122 – In my opinion this sentence is difficult to understand. Please rephrase it. My suggestion is “Presently, the majority of existing studies predominantly focus on surgical metrics, such as a range of motion, complication rates, or strength. However, there appears to be a gap in the literature regarding the effects of surgery on individuals' quality of life.

​Line 122 – 124 – In my opinion this sentence is difficult to understand. Please consider rephrasing it.

​Line 124 – 125 – This sentence is confusing. In my opinion, the authors should consider building upon this conclusion to provide a better explanation about the difference between RTW and return to full duty.

​Line 128 – 129, Line 133 – 134, Lines 135 – 136, Line 137 - 138 – This content is a precise duplication of the information provided in Lines 44-45, 46-47. Kindly consider presenting an alternative sentence that elucidates how your findings contribute to a comparative analysis with existing literature, as well as how these results can contribute to enhanced practices for your readers. Please do not repeat the Introduction section.

​Line 130 – 131 – This information should be reported in the Results section.

Conclusion

​The presented conclusion is clinically relevant and provides important information to readers.

References

​No comment

Figures

​No figures presented

Tables

​Table 1 – Please consider aligning the questions to the left.

​Table 2 – Please explain the abbreviations in the footnotes of the table.

​Table 2 – It is not necessary to write the full name of SST because it was already explained in the Methods section (Line 67). In my opinion by writing the abbreviated version you will also improve the overall design of your table.

​Table 3 – same comment as above.

​Table 4 – Line 103 – Please include abbreviations in table footnotes.

​Table 4 – Please explain what is “Z” and “U”? Did you also report z-scores? If yes, please note that z-scores are best suited when data is normally distributed. Even if they can be extracted from non-normally distributed data, they might not be accurate and meaningful.

​Table 4 – In my opinion this is a case in which min and max values could be very informative for the reader. Sometimes, patients want to be informed about both the best and worst-case scenarios. Therefore, by reporting min and max values for RTW you provide your readers with answers to such questions.

Author Response

Dear Reviewer,

We appreciate your kind review. We did our best to fulfil your expectations, and hopefully managed to correct our paper according to your guidelines. 

Please see our editions:
- we changed some sentences/words to those suggested by you which were more advanced in linguistic terms
- we expanded the introduction so as to discuss in more detail the previous research on this topic, describing in detail the groups studied there; additionally, in the Discussion section, we described the previous RTW results and their interpretation by the authors
- we improved the abstract so that it is not a repetition of the work
- comments regarding the presentation of results were taken into account, we modified the table with MIN/MAX values, additionally answering the question: U and Z are coming from Mann-Whitney U test, but we decided to completely remove it from the table, as they are not really informative for a common reader.
- All technical comments (change of order, development of abbreviations, etc. have been taken into account and applied)

Text that we added or edited has been underlined. According to the journal's instructions, the purpose of the study in the abstract should also be underlined - to distinguish this, it has been additionally bolded. 

We would like to thank you for your kind cooperation.

Kindest regards,

Reviewer 2 Report

It is a very interesting paper.

Finally we know that people treated by ABR can return to work at 9 wks and that the RTW is INDEPENDENT from PROMS or quality of surgery done.

This is a very important info and should be enlighted in the title of the paper.

I suggest you some papers that should be inserted and cited in your paper:

In both of them there are relevant consideration that can enrich your job.

    Arthroscopic remplissage is safe and effective: clinical and magnetic resonance results at a minimum 3 years of follow-up Randelli PS, Compagnoni R, Radaelli S, Gallazzi MB, Tassi A, Menon A Jan 8, 2022 10.1186/s10195-021-00624-5      
    Patient outcomes and return to play after arthroscopic rotator cuff repair in overhead athletes: a systematic review Migliorini F, Asparago G, Cuozzo F, Oliva F, Hildebrand F, Maffulli N Jan 19, 2023 10.1186/s10195-023-00683-w      

It is good enough

Author Response

Dear Reviewer,

We appreciate your kind review. We did our best to fulfil your expectations, and hopefully managed to correct our paper according to your guidelines. 

Please see our editions, all of them are underlined. We also included suggested literature. 
We would like to thank you for your kind cooperation.

Kindest regards,